# Effect of Heat Treatment on the Phase Composition and Corrosion Resistance of 321 SS Welded Joints Produced by a Defocused Laser Beam

**DOI:** 10.3390/ma12223720

**Published:** 2019-11-11

**Authors:** Sergey Vyacheslavovich Kuryntsev

**Affiliations:** Kazan National Research Technical University named after A.N. Tupolev—KAI (KNRTU—KAI), 420111 Kazan, Russia; kuryntsev16@mail.ru; Tel.: +79-872-901-953

**Keywords:** laser beam welding, heat treatment, chemical and phase composition, microstructure, intergranular corrosion resistance

## Abstract

The effect of heat treatment of welded joints made of steel 321 on corrosion resistance, phase composition, residual stresses, and distribution of alloying elements was studied using optical microscope (OM) and scanning electron microscope (SEM), electron dispersive spectroscopy (EDS), X-ray diffraction (XRD), and intergranular corrosion testing (IGC). Samples previously obtained by the authors using defocused laser beam, which led to the formation of directionally crystallized austenite with lathy and skeletal δ-ferrite, were investigated. Based on X-ray diffraction studies in the base metal, the maximum number of peaks of various phases was presented, which decreased after exposure to the heating effect of the welding process and subsequent heat treatment. The distribution of alloying elements, in particular, Ti and Si, was significantly affected by heat treatment depending on the regimes. A spot chemical analysis showed that the nickel content differs in δ-ferrite and austenite by 1.5%–2% whereas the chromium content in these phases is not significantly different. Tests have shown that all samples have high resistance to intergranular corrosion, which can be explained by the insufficient dissolution of titanium carbides in austenite and the absence of chromium carbides formation along austenite grain boundaries, due to high cooling rates when welding by a defocused laser beam, and as a result, the high δ-ferrite content in which chromium dissolves.

## 1. Introduction

In recent decades, austenitic corrosion-resistant steels are a very important group of structural materials used in various industries. Steel 321 is austenitic stainless steel stabilized by titanium from the formation of chromium carbides during welding and other high-temperature treatment processes [1]. This steel has a high corrosion resistance in a number of aggressive liquids, is resistant to intergranular corrosion after welding heating, can be used as a heat-resistant material at temperatures of ~600 °C, while also being highly plastic in deep cold conditions; this steel is used in installations for producing liquid oxygen etc. [2]. These properties are due to the high chromium content (17%–19%) which ensures the metal passivation ability, and the addition of 9%–11% nickel provides the austenitic structure.

This steel grade is widely used in various constructions due to a good complex of operational properties, good weldability, and processability.

The phase composition of the weld metal of austenitic steels depends on the chemical composition (Cr_eq_/Ni_eq_) and cooling rates, especially in some critical temperature ranges [1]. Fusion welding of steels with a given phase composition [3,4] (ASS—austenitic stainless steel, DSS—duplex stainless steel) has an important role in controlling the phase composition of the ferrite content and the content of detrimental phases such as carbides, nitrides, and intermetallic. Also, the cooling rate affects the formation of chromium nitrides which are deposited in ferrite areas because nitrogen is poorly soluble in these phases [5]. The cooling rate plays an important role for the heat affected zone (HAZ) phase composition in which it is difficult to control or modify the phase composition. In some steels, up to several HAZs with different phase compositions can be distinguished [6,7,8]. Moreover, the weld metal can be modified using controlled cooling rates or using filler material [5,9], but this is not possible in the HAZs. This circumstance justifies the full heat treatment of welds of highly sensitive structures.

In addition to controlling the balance of the phase composition, cooling rates are important at certain critical intervals if the cooling rate in the range of 1200–800 °C is too high, then a lot of δ-ferrite and Cr_2_N can be formed, which can reduce the corrosion resistance [1,5]. Arc-welded 321 SS welds are distinguished by a coarse columnar macrostructure, as the weld metal contains a ferritic component due to dendritic segregation [4,10,11].

On the other hand, the content of δ-ferrite (5%–15%) can lead to a decrease in the probability of the appearance of crystallization cracks due to different thermal expansion coefficient of austenite and ferrite (α_γ_ = 17.3, α_δ_ = 9.9 × 10^−6^ m/mK), which differs from the crystallization model of full ferrite or full austenite [12]. Additionally, harmful impurities such as sulfur and phosphorus have a higher solubility in the δ-ferrite, which limits their interdendritic region during primary ferrite solidification [13,14]. The solidification mode of 321 SS is based on the Cr_eq_/Ni_eq_ ratio, and as a rule, this is the FA (ferrite–austenite) mode. Crystallization begins with the formation of initial ferrite and then continues with the formation of austenite [10].

The above aspects of the cooling rate influence on the phase composition predetermine the choice of welding method. As a rule, such methods of fusion welding as submerged-arc welding, arc or plasma welding, and laser and electron-beam welding [4] are most widely used. Furthermore, when it comes to the advantages of laser and electron beam welding, such as penetration depth and high welding speed, lower specific heat input and thermal distortion, narrow weld and HAZs, high precision, ease of automation, etc. [3,15], then in the case of ASS or DSS welding, this is a controversial issue, since in this case high cooling rates lead to a negative effect on the phase balance of the weld metal and HAZ, and, accordingly, on the mechanical and corrosion properties [16,17,18,19]. 

In the case of laser welding of 321 SS, high cooling rates lead to the fact that not all of the δ-ferrite is transformed into austenite [20]. Technological methods such as beam defocusing, wobbling or oscillations (longitudinal, transverse, circular) or the concurrent heating [21,22,23,24,25] are used to reduce the cooling rates during laser welding and electron beam welding.

In a previous paper of the author [20], the positive effect of using a defocused beam in laser welding at a non-zero gap was revealed. However, the use of a defocused laser beam significantly influenced the phase composition of the weld and led to the formation of about 8%–20% of a δ-ferrite. 

Welding by the defocused laser beam or using oscillations [26] is necessary in the case of welding the medium and large thickness sheets. The gap between the edges that are being welded is formed because of the high stiffness of sheets with large thickness. It is necessary either to reduce the gap which is very time consuming or to increase the size of the weld pool by the laser beam oscillation or defocusing, or applying hybrid laser-arc welding or using dual beam welding, etc. These technological methods have a double influence on the behavior of the weld pool and the metallurgical processes in the weld pool. On the one hand, an increase of the weld pool leads to the fact that the molten metal stays in the liquid state longer [27,28], respectively, the rate crystallization decreases, which allows the formed pores to come to the surface [20]. On the other hand, as a result of an increase of the weld pool, volatile alloying elements like chromium, manganese, etc., are burned out [29]. This affects the chemical and phase composition, and therefore the properties of the weld. Also, the crystallization rate affects the formation of hot cracks [14], the presence of internal stresses of the first and second kind, and a decrease in the formation of δ-ferrite.

Since the welded joints of structures operate in contact with aggressive liquids, they must be resistant to IGC [1,2,30,31,32] and other types of corrosion [33,34,35,36,37,38], while maintaining the required level of mechanical properties [39,40]. In the studied 321 SS, intergranular corrosion is based on the formation of chromium carbides along the grain boundaries. For dissolving chromium carbides, stabilizing annealing at 850–900 °C is used, which leads to the formation of harmless titanium carbides. For maximum ductility, annealing is carried out at a higher temperature in the temperature range of 930–1090 °C. Quenching in the temperature range of 930–1090 °С destroys the columnar microstructure of welds and contributes to the partial dissolution of the ferritic component. Besides the chemical composition of steels and welded joints [40,41], various technological factors and technical operations, such as heat treatment [42,43,44], nitrogen content in shielding gas [45,46,47,48], welding regimes [49], and repeated welding heat [50] etc., also significantly affect the corrosion resistance of welded joints metal. On the other hand, heat treatment can lead to the appearance or removal of welding deformations [51,52,53], and an undesirable change in the phase composition [54]. Studies of the influence of two types of heat treatment on the properties of a welded joint have been conducted because the literature does not provide experimental data on the effect of heat treatment on the corrosion resistance of welded joints obtained by the defocused laser beam.

The aim of this work is to study the effect of heat treatment on resistance to intergranular corrosion, residual stresses, phase, and chemical composition of 321 SS welded joints obtained by welding with the defocused laser beam.

## 2. Experimental

### 2.1. Laser Welding Process and Optical Microscopy

321 SS type is steel from high alloyed steels that is well welded both by fusion welding and the resistance welding process. As a rule, heating during welding is not required, as the main problems during welding are solidifying cracking susceptibility and a decrease in corrosion resistance. It is important to maintain the required level of the stabilizing element (Ti) and minimize carbon intrusion into the weld, the content of which should be maintained at the ratio of Ti/C ≤ 7 [2] to stabilize the resistance to intergranular corrosion. In the case of laser welding, as a result of high cooling rates, 5%–15% δ-ferrite is formed which minimizes the appearance of hot cracks that can form during fully austenitic crystallization [5].

In the previous study, the author described the obtaining process of the weld under study [20]. The 321 type stainless steel was chosen for experimental research, the delivery condition, rolling, and the dimensions of workpieces before welding (L × W × T) were 200 × 100 × 5 mm, and after welding 200 × 200 × 5 mm. The chemical composition is shown in Table 1.

Welding was performed on the set of KUKA KR 120 R 2700 extra HA, fiber laser LS-20 of "IPG-Photonics" (Oxford, MA, USA), the welding head LK-690, KUGLER GmbH. The wavelength of the fiber laser was 1070 nm the focal length was 450 mm. Optimal welding regimes were selected experimentally by varying the defocusing distance (+ 10, + 15, + 20 mm) and welding speeds of 20, 25, 30, 35, and 40 mm/s at a constant laser power of 6.5 kW [20]. According to the results of the conducted experiments the following welding regimes were chosen: the defocusing distance was 20 mm, the laser power was 6.5 kW, with a welding speed of 20 mm/sec, and the 99.99% purity argon was used as a shielding gas to protect the top part of the molten pool, with the distance between shielding tubes and the surface of the workpiece at 1.0–1.2 mm. 

After welding, the samples were tested by "North Star Imaging" XView CTX 5000 (NORTH STAR IMAGING, Rogers, MN, USA), 2D-images of welded joints were obtained for determining the presence of pores [20]. The samples were cut on the Buehler AbrasiMatic 300 (BUEHLER LTD, Lake Bluff, IL, USA) equipment (L × W × T—10 × 10 × 5 mm). Marble reagent (HCl—100 sm^3^, CuSO_4_—20 gr, H_2_O—100 sm^3^) was used to identify the microstructure. The microstructure study before and after heat treatment and after IGC testing was carried out by Axiovert Observer.D1m of "Carl Zeiss" microscope (Carl Zeiss, Jena, Germany). In the previous paper, the authors described in detail the study of the influence of welding regimes on the microstructure and phase composition of the weld metal, and welding regimes were picked out to provide a welded joint without pores and cracks see Figure 1 [20]. Figure 1 shows that the weld metal represents an austenitic columnar dendritic structure with a ferritic component resulting from dendritic segregation. The microstructure of the obtained weld is a cast dendritic structure consisting of dendrites along the edges which are directed from the HAZ to the weld center and differently oriented dendrites, which were formed in the center of weld top part. Also, in the previous work, the authors made the quantitative assessment of the austenite/ferrite ratio and the length of directed dendrites depending on the laser welding regimes. The optimal regimes, providing high-quality welded joints (without pores and cracks), led to the formation in the weld top part of the microstructure of dendrites containing 20% δ-ferrite directed from the HAZ to the center of the weld, and with the dendrites oriented differently containing 27.5% δ-ferrite in the weld center.

### 2.2. Heat Treatment

As noted in the introduction section, the need for the heat treatment of the welded joint is justified by the fact that when the weld metal is heated up to 1200–1250 °C titanium carbides dissolve in austenite. When cooling in the critical temperature range (875–450 °C), the solid solution decomposes, the carbide phase falls out along the austenite grain boundaries, and chromium depletes the austenite boundary regions. As a result, chromium carbides are formed along the grain boundaries, this leads to the weld metal susceptibility to intergranular corrosion testing (IGC). For dissolving chromium carbides, stabilizing annealing at 850–900 °C is used, which leads to the formation of harmless titanium carbides. For maximum ductility, annealing is carried out at a higher temperature in the temperature range of 930–1090 °C. Quenching in the temperature range of 928–1093 °С destroys the columnar microstructure of welds and contributes to the partial dissolution of the ferritic component. Based on this data, two modes of heat treatment presented in Table 2 were selected.

The electric furnaces SNOL 7.2/1300 (AB "Umega", Utena, Lithuania) were used to carry out the heat treatment without any protective atmosphere. The dimensions of samples for heat treatment (L × W × T) were 200 × 180 × 5 mm. In order to save time and energy, the sample HT_1 was cooled in a furnace to a temperature of 500 °C (below the critical temperature of a breakdown of solid solution), then it was cooled in water.

### 2.3. Intergranular Corrosion 

IGC testing was carried out in accordance with ASTM A262-Practice E-“The Strauss Test”. Five samples with dimensions of (L × W × T) 120 × 15 × 4 were prepared from each series of samples (NHT, HT_1, HT_2). Before the IGC testing, sensitization heating was carried out at a temperature of 650 ± 10 °C with the heating time of 60 ± 5 min followed by air cooling. Next, the samples were kept in a boiling aqueous solution of copper sulfate and sulfuric acid in the presence of metallic copper (chips) for 15 hours and then bent at three points for 180° over an equal diameter bend. After IGC testing and three-point bending of the samples, transverse microsections were prepared from them. Preparation and etching of microsections were carried out using the same technology as indicated in Section 2.1.

### 2.4. X-Ray Diffraction

An X-ray structural analysis was performed to determine the phase composition of the weld metal before and after heat treatment. For this, a Rigaku SmartLab multifunction diffractometer (Rigaku, Japan) was used. The diffractograms were processed using the PDXL-2 software (Version 2.8.4.0) package and using the ICDD PDF-2 database The analysis of residual stresses was also carried out using the sin^2^(Ψ) method, which was performed using the PDXL-2 software package. The measurements were carried out on the top front surface of the weld, as it had sufficient area for analysis (5 × 5 mm) in comparison with the middle and root parts. Samples were prepared from base metal (BM), as welded joint (NHT), and samples after the heat treatment (HT_1, HT_2). For analysis, the surface of the samples was subjected to additional grinding and polishing using diamond abrasive disk (120, 220, 500 grit) and polishing using diamond emulsions (9 μm, 3 μm, and 1 μm). Then all the samples were etched in a 5% alcoholic solution of nitric acid.

### 2.5. SEM Microstructure and Chemical Composition

The study of the microstructure and chemical analysis were carried out on a Scanning Electron Microscope (SEM) Carl Zeiss Auriga Crossbeam (Carl Zeiss, Jena, Germany). In particular, the microstructure and chemical composition of the transition zones from the base metal to the weld metal were investigated depending on the heat treatment modes. For electron microscopic analysis, SE (secondary electron) regimes were used, for primary visualization of the overall morphology of the samples in panoramic photography. The chemical composition of the samples was determined during the microprobe X-ray spectral analysis based on the Oxford Instruments INCA X-Max energy dispersive spectrometer (EDS) (resolution 127 eV, Carl Zeiss, Jena, Germany). The analysis included the determination of the spot chemical composition of the samples and the construction of spectra using accelerating voltage and current SE1, 20 kV, 700 pA. The studies of the distribution of the chemical elements on a selected area were carried out on the SEM by obtaining the distribution maps of the elements.

## 3. Results and Discussion

### 3.1. Microstructure after Welding

As already mentioned in the Introduction in the previous work, the authors described in detail the study of the influence of welding regimes on the microstructure and phase composition of the weld metal. Welding regimes, providing the welded joint without pores and cracks [20], which is a columnar dendritic structure with a ferritic component resulting from dendritic segregation, were picked out. The regimes under which a high-quality welded joint was obtained including the following parameters the defocusing distance was 20 mm (focal length was 450 mm), with a laser power of 6.5 kW, and a welding speed of 20 mm/sec. According to formulas (1 and 2), for a given steel, Cr_eq_/Ni_eq_ = 1.68, that is, this steel is resistant to hot cracks, taking into account the content of S + P = 0.055.
Cr_eq_ = Cr + 1.37Mo + 1.5Si + 2Nb + 3Ti(1)
Ni_eq_ = Ni + 0.31Mn + 22C + 14.2N + Cu(2)

Additionally, the solidification mode can be determined by the ratio of Cr_eq_/Ni_eq_. In this case (FA): ferritic austenitic 1.48 < Cr_eq_/Ni_eq_ < 1.95, i.e., [9]:*L → L + δ → L + δ + (γ + δ) _per/eut_ → γ + δ*.

As a result, the final microstructure will be a skeletal and/or lathy ferrite resulting from ferrite to austenite transformation.

However, it is impossible to comment exactly about the crystallization mode for such a non-equilibrium and high-speed process like laser welding. According to the microstructure of the top and bottom parts of the weld (Figure 1), the speed and the volume of the weld pool of the top and bottom parts differ significantly [55]. In addition, high cooling rates during laser welding can affect the formation of chromium carbides and solid titanium solution in iron.

### 3.2. Microstructure after Heat Treatment 

Figure 2 shows various cross-sections (BM, top, bottom, HAZ) at a magnification of 200 times of the welded joint after different heat treatment modes (NHT, HT_1, HT_2). As can be seen in Figure 2, HT_1 did not have a high influence on the microstructure of all parts of the welded joint, since it was intended to dissolve chromium carbides and reduce δ-ferrite. These changes cannot be assessed at the microstructure and chemical composition level; this aspect will be discussed later in the SEM section. After HT_2 the microstructure changed significantly both in the base metal and in all parts of the weld. In the top central part, the columnar structure is practically not observed only a hereditary smoothed dendritic structure is visible. Large austenite grains without any columnar structures are observed in the middle and bottom part and in the HAZ. However, in the top part, no large austenite grains which dominate in the structure of the bottom part, the base metal, and the HAZ are observed. Photos of the HAZ were made at the weld top part (top part of the “wine glass” form of laser welding seam) which consist of directed dendrites [20] Figure 1. In Figure 2, no structural heredity of directed dendrites is observed in the HAZ of the HT_2 sample, whereas a residual dendritic structure is observed in the top central part of the same sample. A decrease in the amount of δ-ferrite, which was supposed to be partially dissolved, is observed visually.

Figure 3 shows photographs of the microstructure sections of the top and bottom parts of the weld metal without heat treatment. The microstructure of the top part contains a greater amount of δ-ferrite, and in the bottom, the amount of δ-ferrite is much smaller, and the plates of widmanstatten austenite are observed, which is caused by higher cooling rates of the weld root part. The presence of δ-ferrite is observed at the edge of the weld top part, which affected the formation of microcracks after heat treatment according to the HT_2 mode.

After heat treatment, according to the HT_2 mode, several microcracks are observed in the central top part of the weld (Figure 4a,b). This is caused by different thermal expansion coefficients of γ and δ phases (α_γ_ = 17.3, α_δ_ = 9.9 10-6 m/mK), which at high cooling rates in water lead to the formation of surface cracks. Although the amount of δ-ferrite after austenitization (HT_2) should be minimal. Also, the density of austenite is 2% higher compared to that of ferrite, that is, the volume of austenite is less than ferrite, which could also contribute to a decrease in the volume of the weld and the appearance of cracks. However, these microcracks are not observed in the root of the weld, which is based on a smaller amount of δ-ferrite in the weld bottom part (Figure 3) and a smaller volume of the weld bottom part (Figure 1).

The appearance of these cracks should have a negative impact on the corrosion resistance of the welded joint after heat treatment according to the HT_2 mode. However, in accordance with ASTM A262-Practice E, “the Strauss test”, it is necessary to prepare the surface of the samples mechanically before testing, which led to the removal of the layer with cracks. The work [56] shows the positive effect of mechanical surface treatment on the formation of cracks during stress corrosion tests, thanks to the reduction of surface compressive stresses and the reduction of roughness.

### 3.3. SEM

#### 3.3.1. Chemical Composition

The chemical composition analysis was carried out by means of a microprobe X-ray spectral analysis based on the Oxford Instruments INCA X-Max energy dispersive spectrometer (resolution 127 eV). Considering that the NHT and HT_1 samples retained dendritic microstructure and a strongly pronounced fusion line, a comparative analysis of the point chemical composition of these samples was carried out. Spectra 1 and 2 were taken from austenite HAZ regions, and spectra 3 and 4 from regions of the fusion line (weld metal) Figure 5. In the NHT sample titanium was detected only in spectrum 3 (Ti wt. 0.37%) in δ-ferrite on the fusion line, whereas after heat treatment in the HT_1 sample titanium was detected in all 4 spectra relatively evenly (Ti wt. 0.22–0.36%) (Figure 6). This suggests that titanium is evenly distributed for the formation of harmless carbides. A comparative analysis of the nickel and chromium content in all spectra of the NHT, HT_1 samples was carried out, since the weld metal of these samples had a dendritic structure as compared to the HT_2 sample. In accordance with the fact that nickel is austenite former, a higher amount of nickel was found in spectra 1 and 2 (9.47%–10.82%), a smaller amount in spectra 3 and 4 (7.95%–9.05%) (Table 3). Chromium which is ferrite former was found in spectra 3 and 4 (17.40%–18.61%), and approximately in the same amount in spectra 1 and 2 (17.51%–18.19%) [9]. When analyzing the amount of chromium, it can be stated that it is almost the same in spectra 1 and 2 and spectra 3 and 4, as the difference falls within the limits of measurement error. Whereas the difference in the nickel content in spectra 1, 2 and spectra 3, 4 is about 1.5%, which is the result of a change in the phase composition. Because of the highest cooling rates δ-ferrite which does not have time to transform into austenite depleted in nickel content is formed on the fusion line.

Since after austenitization the microstructure of the HT_2 sample was more homogeneous, and a dendritic structure was not observed on the fusion line as in the NHT and HT_1 samples, the analysis on eight spectra taken from opposite sides of the fusion line was made (Figure 7). Spectra 1 and 2 were taken from the fusion line (FL), and spectra 3 and 4 from the weld metal (WM), and spectra 5–8 from the base metal (BM). In general, in all spectra, the alloying elements are distributed relatively evenly, except for high content of titanium in spectra 1 and 5 in areas with a similar microstructure. Also, spectra 5 and 6 (base metal) in chromium (20.39%–21.21%) and nickel (6.63%–6.97%) content significantly differ from all spectra under consideration, which is probably because these spectra were taken from ferritic areas. In the remaining spectra, the uniform distribution of the main alloying elements corresponding to the chemical composition of the base metal is observed; chromium (18.22%–18.07%), nickel (8.13%–9.40%), titanium (0.23%–0.50%), silicon (0.37%–0.62%) (Table 4).

In the HT_2 sample, titanium is also unevenly distributed and concentrated as in the NHT sample, which is associated with insufficient time for diffusion processes to provide a more uniform distribution of titanium because of high cooling rates during heat treatment and laser welding. This is also confirmed by the point analysis of the chemical composition.

When analyzing the SEM images of the fusion line metal of the HT_2 sample, a weakly pronounced boundary between the base metal and the weld metal is observed. The base metal has a directional macrostructure of austenite along the rolling direction (Figure 8a), in the area of the fusion line there is a needlelike microstructure, similar to quenching metastable phases (Figure 8b). In addition, a multidirectional needle-like structure is observed in the base metal (between austenitic elongated grains) and in the weld metal (between large austenite grains), it can be explained by the formation of quenching phases in areas in which δ-ferrite existed before quenching, for example, in the areas of the fusion line, where the primary crystallization occurred after welding and the δ-ferrite did not have time to transform into austenite. As a rule, in 321 SS austenite is unstable, cooling in the region of negative temperatures or plastic deformation, which can lead to the formation of martensite. In this case, there is plastic deformation (delivery condition before welding is rolled), the formation of a large amount of δ-ferrite 15%–20% after welding in the weld and HAZ, and high cooling rates during quenching in water. Quenching phases can also be observed in areas depleted in nickel and enriched in chromium (as indicated above in Section 3.3.1 Chemical composition, spectra 5 and 6).

When analyzing the EDS mapping, all alloying elements in the studied spectra were distributed relatively evenly with the exception of titanium and silicon. Figure 6 shows that the heat treatment modes significantly influenced on the distribution of titanium and silicon the distribution in the NHT sample is very concentrated titanium is almost in one point silicon is in several points of the area under consideration. Titanium is distributed as evenly as possible in the HT_1 sample, silicon in this sample is also uniformly distributed with the exception of the fusion line and some sections of δ-ferrite. In the HT_2 sample, the silicon is even more evenly distributed relative to the HT_1 sample, but not so intense. In the HT_2 sample, the titanium is also uniformly distributed, but not as intense as in the sample HT_1.

#### 3.3.2. SEM Microstructure

Since the cast microstructure of the weld metal was kept in the NHT and HT_1 samples, a comparative analysis only of these samples was carried out. In Figure 9a,b, a significant decrease of the dark areas containing δ-ferrite in the weld metal of HT_1 sample compared to the NHT sample is observed. The transition area from the base metal to the weld metal in sample HT_1 is smoother, which is due to the enlargement of austenite fine particles and the dissolution of δ-ferrite (Figure 9c,d). In the weld metal of the NHT sample which is close to the fusion zone consisting of directed dendrites [20] the widmanstatten austenite, which is the same as in the weld root, is observed (Figure 3). However, in contrast to welded joints produced by arc welding [10,11] the crystallization rate of which is much lower, the fusion line of the base metal and weld metal is sharper and contains δ-ferrite along the boundaries of small austenite dendrites (Figure 9e). When comparing the microstructure images of the fusion line of the NHT and HT_1 samples, it can be seen that the primary crystallization zone of the NHT sample contains small austenite dendrites in the δ-ferrite matrix (Figure 9e), while in the HT_1 sample small austenite dendrites have enlarged (Figure 9f) as the result of their coagulation under the influence of heat treatment.

### 3.4. X-Ray Diffraction

#### 3.4.1. Phase Composition

An XRD structural analysis was performed to determine the qualitative phase composition. The measurements were carried out on the top front surface of the weld, as it had sufficient area for analysis (5 × 5 mm) in comparison with the middle and root parts of the weld (Figure 1). Phase analysis of the base metal (BM), the weld without heat treatment (NHT), and the weld with heat treatment (HT_1, HT_2) was carried out. The phase composition of these samples is represented by a matrix of austenite (Fm-3m (225), ICDD PDF-2 00-031-0619), and δ-ferrite, and as a secondary phase (Im-3m (229), ICDD PDF-2 00-006-0696). From the obtained data, it can be seen that the peaks 200 (δ), 220 (δ), 310 (δ), and 400 (γ) were not found in sample HT_1 (Figure 10). Peaks 400 (γ) are present only in samples HT_2, BM, and 310 (δ) only in the samples NHT, BM. This shows the effect of heat treatment on the qualitative phase composition. In the sample of the base metal BM, 11 different peaks are indicated, while in the rest of the samples 1 or 3 peaks are not observed, as indicated above. It also indicates that the phase composition is influenced not only by the heat treatment but also by the welding thermal cycle (NHT sample does not contain the peak 400 (γ), in contrast to the HT_2 sample). The largest number of different peaks is represented in the sample of the BM in comparison with the weld metal after welding and after heat treatment. It demonstrates a more homogeneous phase composition of the HT_1 sample.

#### 3.4.2. Residual Stress Measurement 

The analysis of residual stresses was carried out using the sin^2^ (Ψ)-method, in which the diffraction patterns at several inclination angles of the sample relative to the incident X-ray beam were recorded. The residual stresses were calculated according to the formula:(3)$$\sigma =\;-\frac{E}{2\left(1+\nu \right)}\times ctg\theta_{0}\times \frac{\pi }{180}\times \frac{\Delta \left(2\theta \right)}{\Delta \left(sin^{2}\Psi \right)}$$
where *E* is the elasticity modulus (Young’s modulus); 

*ν* is the Poisson’s ratio;

*θ_0_* is the angular position of the sample’s peak, which has no residual stresses (BM);

$\frac{\Delta \left(2\theta \right)}{\Delta \left(sin^{2}\Psi \right)}$—the linear graph slope of the dependence of the diffraction peak position on the sine of inclination ψ.

The following parameters characteristic for 321 SS were used in the calculation: E = 1.98 10^-5^ MPa, ν = 0.27. The γ (311) peak was investigated since, when recording a sample in the area with this peak at different ψ angles, the peak shifts are most pronounced. The angular position of the peak 311 *θ_0_* was determined from the diffraction pattern of the sample No. 1, and it was *θ_0_* = 90,61°. When obtaining the diffractograms used to determine the residual stresses, the incident X-rays were directed under different angles relative to the weld, namely: longitudinal (0°), diagonal (45°), and transverse direction (90°), Table 5. As a rule, the residual welding stresses of 321 SS are comparable with the yield stress (250–300 MPa), the longitudinal and transverse stresses are approximately equal after arc welding, and the stresses along the Z-axis deep into the weld are much smaller. In this case, in the NHT sample, the longitudinal and transverse stresses are approximately equal to each other (−265 ± 133 MPa, −240 ± 71 MPa, respectively). As can be seen from Figure 1, there is no segregation of the weld metal, the volume of the weld metal increased (no filler material was used), the upper part is flat, and sagging of weld metal is observed at the root. Wherefrom, it can be assumed that the volume of the weld pool increased due to the formation of δ-ferrite (20%–25%), which reduces the transverse compressive stresses and could lead to the appearance of tensile stresses. Of course, for an unambiguous statement, the stresses of the middle and root parts of the weld should be measured. However, in this paper, other influencing factors were considered. It should also be noted that in [20], welding was carried out with a gap formed by edges with the use of a laser cut (this is the reason for using the defocused laser beam), which also leads to a decrease in transverse welding stresses.

The HT_2 sample has the greatest longitudinal (−298 ± 24 MPa) and transverse (−368 ± 87 MPa) residual stresses, which is associated with the appearance of stresses caused by different thermal expansion coefficients γ and δ on the weld surface after cooling in water, and the appearance of cracks (Figure 4). As studies in the previous paper [20] show, a large amount of δ-ferrite (20%–25%) with a coefficient of thermal expansion almost two times less than that of austenite (α_γ_ = 17.3, α_δ_ = 9.9×10^−6^ m/mK) is formed on the surface of a weld produced by a defocused laser beam.

The HT_1 sample has the longitudinal (−228 ± 87 MPa) and transverse (−359 ± 81 MPa) stresses, that is, the longitudinal stresses are comparable to that of the NHT sample, which is associated with a low cooling rate up to 500 °C, and the transverse stresses to that of the HT_2 sample, which is associated with insufficient temperature for complete austenitization.

### 3.5. Intergranular Corrosion Resistance

Intergranular corrosion (IGC) testing was carried out according to ASTM A262-Practice E, “the Strauss test”. From each series of samples (NHT, HT_1, HT_2), five samples were prepared, which after IGC testing were subjected to three point bending and the analysis of transverse sections (Figure 11a–c).

After carrying out three-point bending with a load, no microcracks were observed visually or with a magnifying glass on the surface from the weld root side. The microstructure analysis also did not show the presence of cracks or microdamages along the boundaries of dendrites (NHT, HT_1samples) or grains (HT_2 sample). Even though the HT_2 samples had surface cracks after heat treatment, they were removed during machining before IGC testing, as indicated in Section 3.2.

## 4. Conclusions

The carried out studies of the heat treatment effect on the microstructure, chemical and phase composition, and corrosion resistance of the welded joint produced by the defocused laser beam showed the ambiguous effect of heat treatment. Welded joint metal without heat treatment (NHT) like the rest samples HT_1 and HT_2 showed high resistance to intergranular corrosion. The samples showed no cracks after testing and three-point bending, neither on the weld surface nor on the weld cross-section under an optical microscope. It can be explained by the insufficient dissolution of titanium carbides in austenite and absence of chromium carbides formation along austenite grain boundaries due to high cooling rates when welding by a defocused beam and as a result, the high δ-ferrite content in which chromium dissolves. Also, the small size of the weld pool influenced the minimization of residual stresses in the as-welded joint.

Based on the carried out research, the following qualitative and quantitative conclusions can be made:

(1) NHT sample has the minimal compressive longitudinal (−265 ± 133 MPa) and transverse (−240 ± 71 MPa) stresses.

(2) Based on the EDS analysis of the fusion line, the distribution of the chromium and nickel— the main alloying elements of 321 SS—in the dendritic region of austenite and the interdendritic region of δ-ferrite of NHT and HT_1 samples differs by 1.5%–2% in nickel content. In HT_2 sample, the content of chromium and nickel differs by 2%–2.8% in various points.

(3) On the fusion line, the distribution of such alloying elements as Ti and Si changed significantly after heat treatment, the most uniform distribution of titanium was recorded after HT_1 and silicon after HT_2.

(4) Analyzing the obtained experimental data, it is possible to conclude that from the point of view of resistance against IGC heat treatment of the welded joint, the results obtained by the defocused laser beam indicate that is not necessary to carry out. Nevertheless, from the point of view of uniform distribution of alloying elements the most suitable heat treatment mode is the HT_1.

## Figures and Tables

**Figure 1 materials-12-03720-f001:**
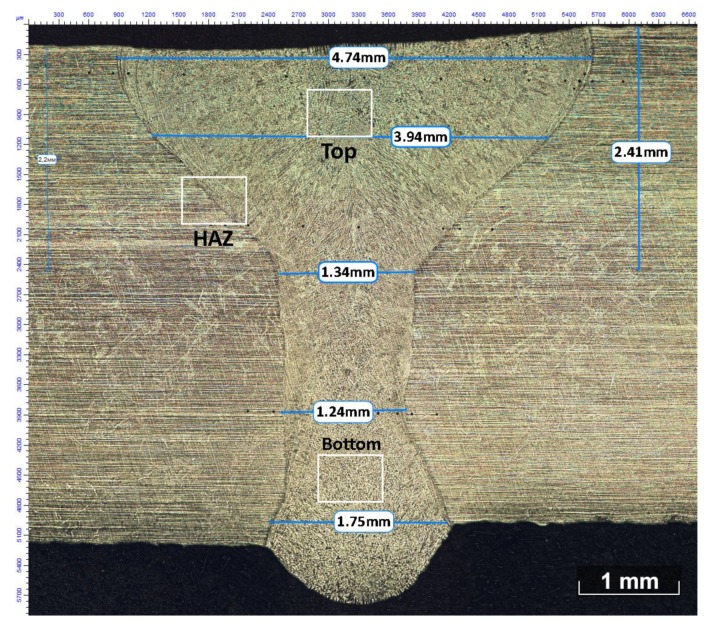
Macrostructure of as-welded sample (NHT – No Heat Treatment); white squares–zones shown in a next figure.

**Figure 2 materials-12-03720-f002:**
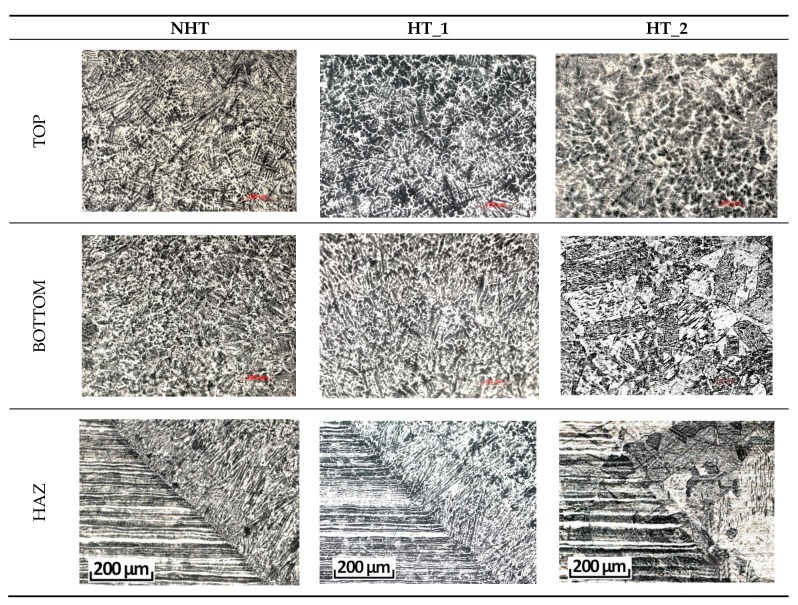
Cross-sections (Top, Bottom, HAZ) at a magnification of 200 times. BM, base metal; HAZ, heat affected zone.

**Figure 3 materials-12-03720-f003:**
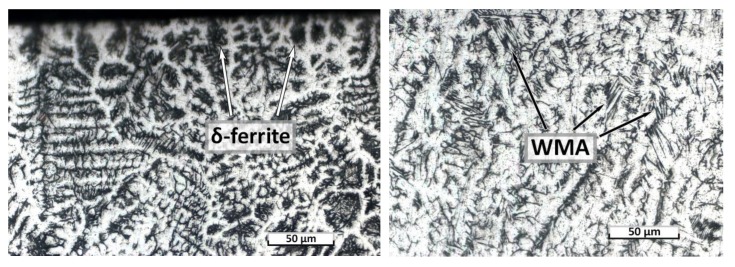
Microstructure of the top and bottom parts of the weld metal of a NHT sample.

**Figure 4 materials-12-03720-f004:**
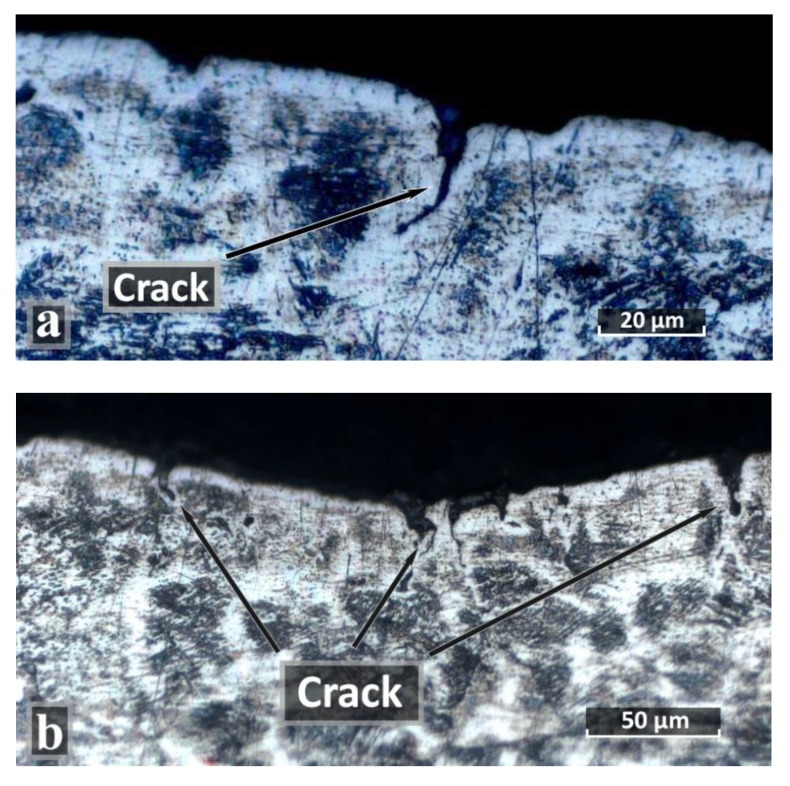
Microcracks in the central top part of the weld of a HT_2 sample. (**a**) magnification of 1000 times; (**b**) magnification of 500 times.

**Figure 5 materials-12-03720-f005:**
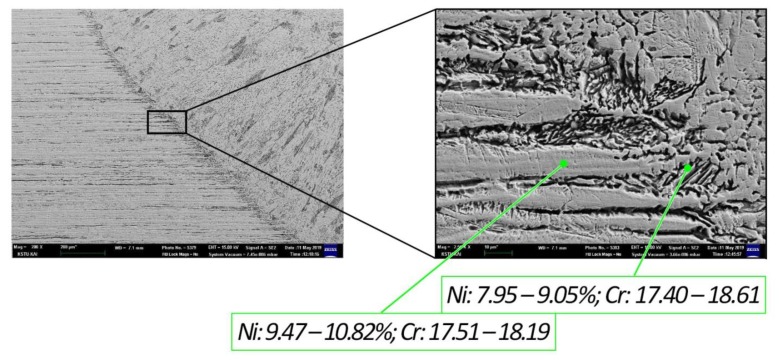
SEM images of a NHT sample.

**Figure 6 materials-12-03720-f006:**
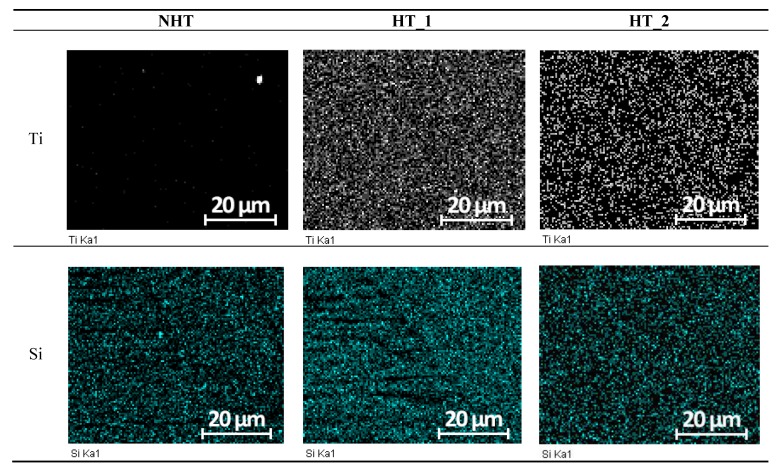
Electron dispersive spectroscopy (EDS) of the titanium and silicon.

**Figure 7 materials-12-03720-f007:**
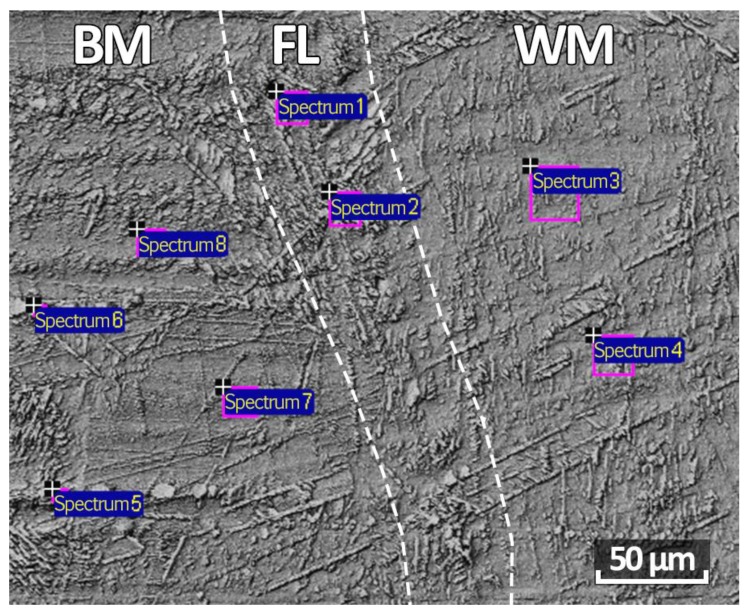
Points of the spectra of a HT_2 sample.

**Figure 8 materials-12-03720-f008:**
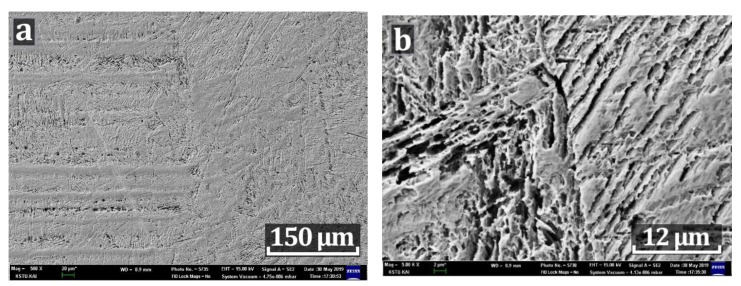
SEM images of a HT_2 sample. (**a**) magnification of 50 times; (**b**) magnification of 500 times.

**Figure 9 materials-12-03720-f009:**
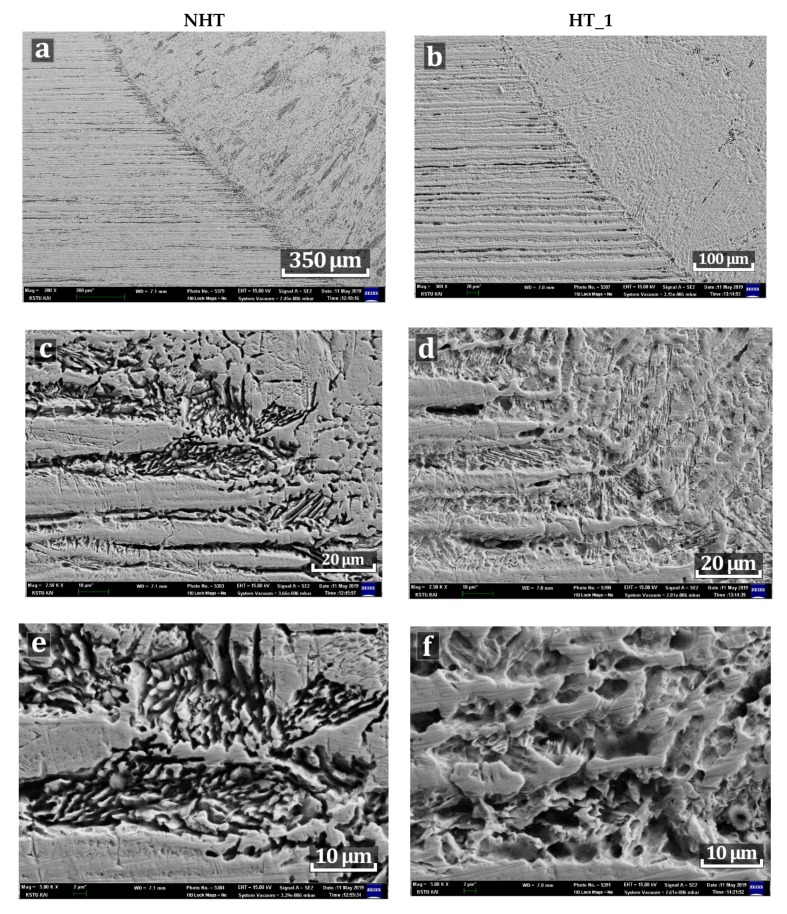
SEM images of the NHT and HT_1 samples. (**a**) magnification of 50 times of a NHT sample, (**b**) magnification of 50 times of a HT_1 sample, (**c**) magnification of 250 times of a NHT sample, (**d**) magnification of 250 times of a HT_1 sample, (**e**) magnification of 500 times of a NHT sample, (**f**) magnification of 500 times of a HT_1 sample.

**Figure 10 materials-12-03720-f010:**
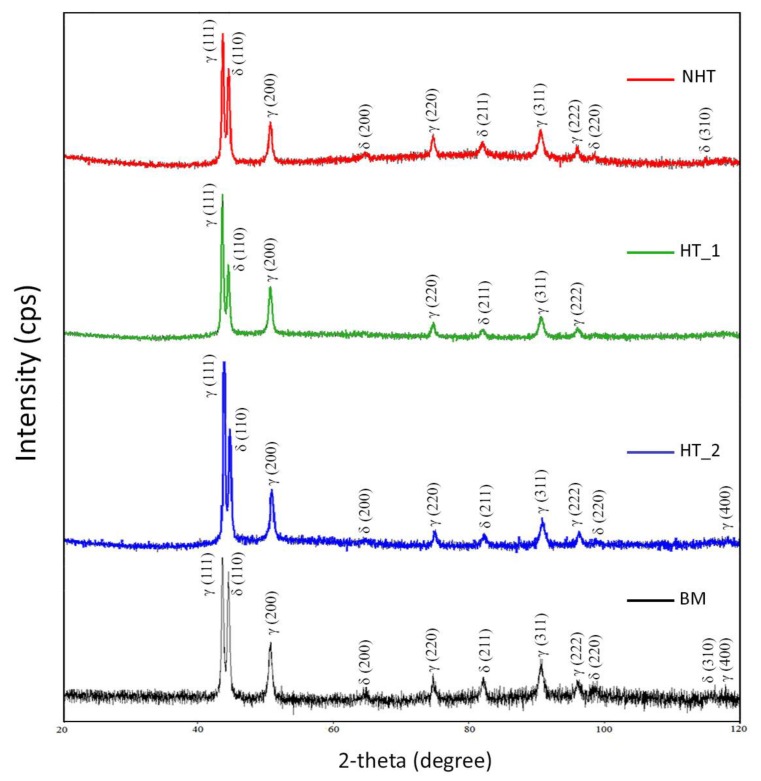
XRD peaks of the NHT, HT_1, HT_2, and BM samples.

**Figure 11 materials-12-03720-f011:**
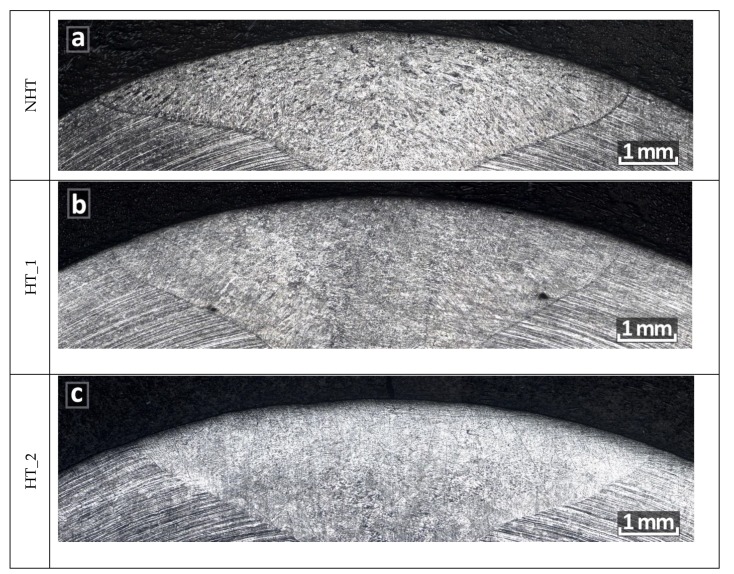
Transverse sections after intergranular corrosion testing (IGC) testing and three point bend test (**a**) NHT sample; (**b**) HT_1 sample; (**c**) HT_2 sample.

**Table 1 materials-12-03720-t001:** Chemical composition of a specimen, wt %.

C	Si	Mn	S	P	Cr	Ni	Ti	As	Cu
<0.12	<0.80	<2.00	<0.020	<0.035	17.0–19.0	9.0–11.0	<1.00	–	–

**Table 2 materials-12-03720-t002:** Heat Treatment modes.

Sample	NHT	HT_1	HT_2
Temperature, °C	–	880	1080
Treatment time, min	–	120	120
Cooling area	–	Furnace, until 500 °C, than water	Water

**Table 3 materials-12-03720-t003:** Content of chemical elements in different spectra of NHT and HT_1 samples.

	Cr, wt%		Ni, wt%		Ti, wt%		Si, wt%	
	NHT	HT_1	NHT	HT_1	NHT	HT_1	NHT	HT_1
Spectrum 1	17.86	17.93	10.30	10.69	–	0.22	0.58	0.47
Spectrum 2	18.19	17.51	9.47	10.82	–	0.25	0.52	0.53
Spectrum 3	17.40	18.36	8.74	9.03	0.37	0.35	0.66	0.56
Spectrum 4	18.47	18.61	7.95	9.05	–	0.36	0.53	0.56

**Table 4 materials-12-03720-t004:** Content of chemical elements in different spectra of HT_2 sample.

	Cr, wt%	Ni, wt%	Ti, wt%	Si, wt%
Spectrum 1	18.50	8.40	2.74	0.48
Spectrum 2	18.97	8.13	0.43	0.54
Spectrum 3	18.22	9.40	0.32	0.48
Spectrum 4	18.77	9.31	0.40	0.62
Spectrum 1	20.39	6.97	2.03	0.41
Spectrum 2	21.21	6.63	0.50	0.37
Spectrum 3	18.80	9.22	0.26	0.57
Spectrum 4	18.69	8.98	0.23	0.46

**Table 5 materials-12-03720-t005:** Residual stress measurement.

Sample	Residual Stress *σ*, MPa		
	0°	45°	90°
NHT	−265 ± 133	−363 ± 53	−240 ± 71
HT_1	−228 ± 87	−368 ± 81	−359 ± 81
HT_2	−298 ± 24	−281 ± 26	−368 ± 87

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
