# Peer review of "Effect of Heat Treatment on the Phase Composition and Corrosion Resistance of 321 SS Welded Joints Produced by a Defocused Laser Beam"

_materials, 2019, doi:10.3390/ma12223720_

Round 1
Reviewer 1 Report
The effect of heat treatment on the phase composition and corrosion resistance of welded joints produced by a defocused laser beam included in the reviewed article is very important. In this context presented work can be evaluated as important to the field of material science.
The literature background given in the introduction is wide and detail. The author refers to actual sources and provides proper conclusions based on this analysis. In some places, the paper is too much of a research report than original scientific research.
The main problem with paper is the presentation of results.
Many comments related to the quality of drawings e.g.:
- In fig. 1 there is no description of the zones specified in fig. 2. The font size in the drawing is too small - A general remark to the paper is the incorrect font size in drawings and descriptions.
- There are no descriptions of columns and rows in Fig. 2,6,8,9,11
- There is no description of the photos in Figures 3 and 4 (a, b)
- In figure 7, the zones from the description should be described. Measuring points should only be marked by a number.
- There is no tabular summary of the information contained in the discussion.
In general, the presentation of results and discussions should be improved.
Author Response
Dear reviewer, the author would like to thank you for your comments, which contributed to improving the quality of the manuscript.
Response to Reviewer 1 Comments
The effect of heat treatment on the phase composition and corrosion resistance of welded joints produced by a defocused laser beam included in the reviewed article is very important. In this context presented work can be evaluated as important to the field of material science.
The literature background given in the introduction is wide and detail. The author refers to actual sources and provides proper conclusions based on this analysis. In some places, the paper is too much of a research report than original scientific research.
The main problem with paper is the presentation of results.
Many comments related to the quality of drawings e.g.:
Point 1:
- In fig. 1 there is no description of the zones specified in fig. 2. The font size in the drawing is too small - A general remark to the paper is the incorrect font size in drawings and descriptions.
Response 1: The zones shown in Fig. 2 are specified in Fig. 1. The font size in drawing is increased.
Point 2:
- There are no descriptions of columns and rows in Fig. 2, 6, 8, 9, 11.
Response 2:
Perhaps in the PDF version, which was provided to the Reviewer was not visible columns and rows, in the original version of Word is columns and rows in all Figures 2, 6, 8, 9, 11.
Point 3:
- There is no description of the photos in Figures 3 and 4 (a, b)
Response 3:
Lines 245 – 250, 260 contain a description of Figure 3. Line 254 contains a reference and description of Figure 4.
Point 4:
- In figure 7, the zones from the description should be described. Measuring points should only be marked by a number.
Response 4:
Figure 7 shows the zones from which the spectra were taken (BM, FL, WM). The measuring points are marked by Spectrum numbers.
Point 5:
- There is no tabular summary of the information contained in the discussion.
In general, the presentation of results and discussions should be improved.
Response 5:
Tables 3 and 4 have been added to improve the presentation of results. Table 3 shows the values of the chemical element content in different spectra of NHT and HT_1 samples. Table 4 contains the values of the chemical element content in different spectra of the HT_2 sample.
Table 3: Content of chemical elements in different spectra of NHT and HT_1 samples.
|
|
Cr, wt % |
Ni, wt % |
Ti, wt % |
Si, wt % |
||||
|
NHT |
HT_1 |
NHT |
HT_1 |
NHT |
HT_1 |
NHT |
HT_1 |
|
|
Spectrum 1 |
17.86 |
17.93 |
10.30 |
10.69 |
- |
0.22 |
0.58 |
0.47 |
|
Spectrum 2 |
18.19 |
17.51 |
9.47 |
10.82 |
- |
0.25 |
0.52 |
0.53 |
|
Spectrum 3 |
17.40 |
18.36 |
8.74 |
9.03 |
0.37 |
0.35 |
0.66 |
0.56 |
|
Spectrum 4 |
18.47 |
18.61 |
7.95 |
9.05 |
- |
0.36 |
0.53 |
0.56 |
Table 4: Content of chemical elements in different spectra of HT_2 sample.
|
|
Cr, wt % |
Ni, wt % |
Ti, wt % |
Si, wt % |
|
Spectrum 1 |
18.50 |
8.40 |
2.74 |
0.48 |
|
Spectrum 2 |
18.97 |
8.13 |
0.43 |
0.54 |
|
Spectrum 3 |
18.22 |
9.40 |
0.32 |
0.48 |
|
Spectrum 4 |
18.77 |
9.31 |
0.40 |
0.62 |
|
Spectrum 1 |
20.39 |
6.97 |
2.03 |
0.41 |
|
Spectrum 2 |
21.21 |
6.63 |
0.50 |
0.37 |
|
Spectrum 3 |
18.80 |
9.22 |
0.26 |
0.57 |
|
Spectrum 4 |
18.69 |
8.98 |
0.23 |
0.46 |

Reviewer 2 Report
The paper describes the effect of heat treatments on welded stainless steel. The purpose of the paper appears reasonable, but the paper needs some revision.
In Fig. 1, I can not see the captions. Please increase it. P. 5, please spell out the IGC. in Fig. 2, Please replace top three figures. Make a table for your composition analysis. Fig. 7, please make it English captions. In conclusion, please remove No. 1. I do not agree with No. 1. In conclusion, for potential readers, please tell what you learn with reasons. For example, did you mean that HT 1 is good or necessary?
Author Response
Dear Reviewer, the author would like to thank you for your comments, which contributed to improving the quality of the manuscript.
Response to Reviewer 2 Comments
The paper describes the effect of heat treatments on welded stainless steel. The purpose of the paper appears reasonable, but the paper needs some revision.
Point 1
In Fig. 1, I can not see the captions. Please increase it.
Response 1
The captions are magnified.
Point 2
5, please spell out the IGC.
Response 2
IGC (intergranular corrosion) is spell out in abstract, line 11.
Point 3
in Fig. 2, Please replace top three figures. Make a table for your composition analysis.
Response 3
The top three photos are replaced from Fig. 2.
Tables 3 and 4 are added, containing the results of chemical analysis.
Point 4
Fig. 7, please make it English captions.
Response 4
English captions are added in Fig. 7.
Point 5
In conclusion, please remove No. 1. I do not agree with No. 1.
Response 5
No. 1 in conclusion are remove.
Point 6
In conclusion, for potential readers, please tell what you learn with reasons. For example, did you mean that HT 1 is good or necessary?
Response 6
This comment from the Reviewer is very important for potential readers.
The No. 4 in the conclusions is the following information.
4 Analyzing the obtained experimental data, it is possible to conclude that from the point of view of resistance against IGC heat treatment of the welded joint, obtained by defocused laser beam is not necessary to carry out. Nevertheless, from the point of view of uniform distribution of alloying elements the most suitable heat treatment mode is the HT_1.

Reviewer 3 Report
The authors studied the effect of heat treatment of welded joints made of steel 321 in terms of corrosion resistance, phase composition, residual stresses and distribution of alloying elements. In the introduction, the typical problems of the process are mentioned with an adequate bibliographical analysis.
The reviewer gives to the authors some suggestions:
Point 1: Authors should provide a more accurate description of the laser device characteristics and of the experimental laser welding procedure; Point 2: How many specimen for welding test were used? How many replication of the different scenarios were made? It is opinion of the reviewer that a table summarizing the experimental conditions and the replication could be helpful; Point 3: Results are well presented, but quality of figure 2 should be improved; Point 4:The scale labels are missing in figures 6, 7 and 11.
Author Response
Dear Reviewer, the author would like to thank you for your comments, which contributed to improving the quality of the manuscript.
Response to Reviewer 3 Comments
The authors studied the effect of heat treatment of welded joints made of steel 321 in terms of corrosion resistance, phase composition, residual stresses and distribution of alloying elements. In the introduction, the typical problems of the process are mentioned with an adequate bibliographical analysis.
The reviewer gives to the authors some suggestions:
Point 1: Authors should provide a more accurate description of the laser device characteristics and of the experimental laser welding procedure;
Response 1
The main description of the laser device is given in the manuscript. The laser spot diameter is not indicated, it was 200 microns for the focused laser beam, but it changed with the change of the focal length, which was experimentally measured and described in detail in the author's paper Kuryntsev S.V., Gilmutdinov A. Kh. Welding of stainless steel using defocused laser beam. Journal of Constructional Steel Research 2015, 114, 305 – 313. The focal length was changed in increments of 5 mm (+10, +15, +20 mm), the welding speed was 20, 25, 30, 35, 40 mm/s, with a laser power of 6.5 kW, which were mentioned in current manuscript (Lines 128 – 131).
Point 2: How many specimen for welding test were used? How many replication of the different scenarios were made? It is opinion of the reviewer that a table summarizing the experimental conditions and the replication could be helpful;
Response 2
For the welding test, 3 welded joints were prepared with dimensions of (L × W × T) 200 × 200 × 5 mm (Line 124), of which samples for heat treatment were prepared with dimensions of 200 × 180 × 5 mm (Line 167). After the heat treatment, samples for intergranular corrosion testing were prepared with dimensions 120×15×4 mm (Line 173), samples for metallographic analysis (3 for each heat treatment mode) with dimensions 10×10×5 mm (Line 137), samples for X-ray analysis (1 for each heat treatment mode) with dimensions 5×5×5 mm (Line 187). All these data are given in the text of the manuscript and in Table 2.
Point 3: Results are well presented, but quality of figure 2 should be improved;
Response 3
The quality of Fig. 2 has been improved as much as possible, the line with photos of the base metal was removed by the request of one of the Reviewers.
Point 4: The scale labels are missing in figures 6, 7 and 11.
Response 4
The scale labels are added in Fig. 6, 7 and 11.

Round 2
Reviewer 1 Report
The author took into account the comments of the reviewer.
Reviewer 3 Report
The authors revised the paper according to the reviewer comments.
Paper can be accepted in present forms.